# High Diversity of *Giardia duodenalis* Assemblages and Sub-Assemblages in Asymptomatic School Children in Ibadan, Nigeria

**DOI:** 10.3390/tropicalmed8030152

**Published:** 2023-02-28

**Authors:** Muyideen K. Tijani, Pamela C. Köster, Isabel Guadano-Procesi, Imo S. George, Elizabeth Abodunrin, Adedamola Adeola, Alejandro Dashti, Begoña Bailo, David González-Barrio, David Carmena

**Affiliations:** 1Cellular Parasitology Programme, Cell Biology and Genetics Unit, Department of Zoology, University of Ibadan, Ibadan 200284, Nigeria; 2Division of Clinical Chemistry and Pharmacology, Department of Laboratory Medicine, Lund University, Skåne University Hospital, 221 85 Lund, Sweden; 3Parasitology Reference and Research Laboratory, Spanish National Centre for Microbiology, Health Institute Carlos III, 28220 Madrid, Spain; 4Department of Clinical Sciences and Translational Medicine, Faculty of Medicine, University of Rome “Tor Vergata”, 00133 Rome, Italy; 5PhD Program in Evolutionary Biology and Ecology, Department of Biology, University of Rome “Tor Vergata”, 00133 Rome, Italy; 6Center for Biomedical Research Network (CIBER) in Infectious Diseases, Health Institute Carlos III, 28029 Madrid, Spain

**Keywords:** intestinal parasites, giardiasis, transmission, diarrhea, epidemiology, Africa

## Abstract

*Giardia duodenalis* is a significant contributor to the burden of diarrheal disease in sub-Saharan Africa. This study assesses the occurrence and molecular diversity of *G. duodenalis* and other intestinal parasites in apparently healthy children (*n* = 311) in Ibadan, Nigeria. Microscopy was used as a screening method and PCR and Sanger sequencing as confirmatory and genotyping methods, respectively. Haplotype analyses were performed to examine associations between genetic variants and epidemiological variables. At microscopy examination, *G. duodenalis* was the most prevalent parasite found (29.3%, 91/311; 95% CI: 24.3–34.7), followed by *Entamoeba* spp. (18.7%, 58/311; 14.5–23.4), *Ascaris lumbricoides* (1.3%, 4/311; 0.4–3.3), and *Taenia* sp. (0.3%, 1/311; 0.01–1.8). qPCR confirmed the presence of *G. duodenalis* in 76.9% (70/91) of the microscopy-positive samples. Of them, 65.9% (60/91) were successfully genotyped. Assemblage B (68.3%, 41/60) was more prevalent than assemblage A (28.3%, 17/60). Mixed A + B infections were identified in two samples (3.3%, 2/60). These facts, together with the absence of animal-adapted assemblages, suggest that human transmission of giardiasis was primarily anthroponotic. Efforts to control *G. duodenalis* (and other fecal-orally transmitted pathogens) should focus on providing safe drinking water and improving sanitation and personal hygiene practices.

## 1. Introduction

*Giardia duodenalis* (synonyms *G. intestinalis*, *G. lamblia*) is the etiological agent of giardiasis, a major contributor to the burden of diarrheal disease causing near 190 million symptomatic infections per year globally, adding up to ∼171,100 daily-adjusted life years lost [1]. Clinical infection ranges from asymptomatic carrier state to severe abdominal pain, diarrhea, vomiting, flatulence, malabsorption, anorexia, and weight loss [2]. Long-term sequelae including postinfectious irritable bowel disease and chronic fatigue have also been reported [3]. Despite its medical importance, no human vaccine is available against giardiasis [4], and clinical resistance against 5-nitroimidazoles (the most used drugs for first-line treatment) occurs in up to 50% of cases [5]. Transmission is via the fecal–oral route, either by ingestion of water or food contaminated with cysts of the parasite, or through direct contact with infected humans, animals, or fomites [6].

Giardiasis disproportionally affects young children, many of whom are undernourished, living in limited-resource settings without or with inadequate access to clean water or good sanitation [7]. In endemic settings, giardiasis has been consistently associated with childhood growth faltering, failure to thrive, and impaired cognitive development [8,9]. Findings from prospective longitudinal cohort studies conducted in sub-Saharan African countries indicate that *Giardia* was not an independent risk factor for diarrhea in children [10,11,12]. Integrating asymptomatic carriage of *Giardia* with its role as a diarrhea-causing agent in these settings is, therefore, challenging and controversial [13]. Several factors have been proposed to contribute to the outcome of the infection, including nutritional [14], microbial [15], metabolic [16], and pathogen-strain [17] variables.

*Giardia duodenalis* is now known as a multispecies complex comprising eight (A–H) genotypically distinct assemblages [18]. Most human *Giardia* infections are due to assemblages A or B, although human cases by animal-adapted assemblages C-F are less frequently reported [18,19]. Several molecular-based surveys have attempted to correlate the presence of diarrhea with the genotype of *G. duodenalis*. Large prospective case-control studies conducted in Bangladesh and Mozambique generated contradictory results [20,21], highlighting the need for additional studies aiming at investigating the molecular diversity of the parasite in symptomatic and asymptomatic individuals of all age groups.

In Nigeria, the annual number of deaths from diarrheal diseases per 100,000 people decreased from 275.3 in 1990 to 92.5 in 2019 [22]. Although *Giardia* infection is not usually associated with mortality, giardiasis seems widely present in the country, affecting both asymptomatic and clinical individuals and livestock (Table 1) [23,24,25,26,27,28,29,30,31,32,33,34,35,36,37,38,39,40,41,42,43,44,45,46,47,48,49,50]. In addition, very little is currently known on the genetic variability of *G. duodenalis* circulating in Nigerian human populations. This study aims at investigating the occurrence and molecular diversity of *G. duodenalis* in school children in Ibadan, a city in the Southwestern part of Nigeria.

## 2. Materials and Methods

### 2.1. Study Design and Participants

In this two-point cross-sectional study, individual stool samples were collected from asymptomatic school children aged 5–17 from four districts in Ibadan metropolis: Boluwaji, Moniya, Orogun, and Yemetu. Ibadan is the capital and most populous city of Oyo State in Southwestern Nigeria. It is the third largest city by population in the country after Lagos and Kano, with a total population of 3,649,000 as of 2021. Ibadan has a tropical savanna climate, with an average annual temperature of 25.9 °C and an annual precipitation of about 1467 mm, mostly concentrated in the period August–October. The warmest month is February (average temperature: 28.9 °C) and the coolest month is August (average temperature: 24.4 °C). This study was carried out between August–December 2017 and May–July 2018.

### 2.2. Stool Sample Collection

Informative meetings were held at scheduled times with school pupils on the causes, symptoms, and mode of transmission of giardiasis. Those willing to participate in this study were provided with transparent, sterile stool recipients containing no preservatives and instructed on how to collect a stool sample in a simple and safe way. The stool recipients were labelled with individual numeric codes allocated to each participating child for easy identification of samples. Collected fresh stool samples were placed on ice and transported within one hour to the Cellular Parasitology Laboratory, Department of Zoology, University of Ibadan, for further analysis.

### 2.3. Epidemiological Questionnaires

Individual standardized questionnaires in Yoruba and English languages were completed by a member of our research team in face-to-face interviews with each participating child at sample collection. Questions included demographics, hand and vegetable washing, contact with domestic animals and livestock, source of drinking water, use of treated water, latrine use, and garbage disposal. The information obtained was entered into an Excel spreadsheet (Microsoft Corporation, Redmond, WA, USA) and used for risk association analyses.

### 2.4. Microscopy

Freshly collected stool samples were microscopically examined within 24 h of collection. The formol-ether concentration technique was used [51]. Briefly, 1–2 g of the fecal sample was homogenized in 7 mL of 10% formalin and sieved through surgical gauze as a mechanical filter to remove fecal debris. The sieved suspension was transferred into a clean 15 mL centrifuge tube and 3 mL of ethyl acetate were added. After vigorous shaking, the mixture was centrifuged for 3 min at 3000 rpm. The supernatant was carefully discarded, and fecal smears made from the sediment. Smears were then stained with 1% Lugol’s iodine and examined at 200× for screening and 400× for confirmation of *Giardia* cysts. Each slide was examined by two independent microscopists. Smears yielding discrepant results were resolved by a third microscopist.

### 2.5. DNA Extraction and Purification

Genomic DNA was isolated from about 200 mg of fecal samples yielding positive/discrepant results for *G. duodenalis* at microscopy examination by using the QIAamp DNA Stool Mini Kit (Qiagen, Hilden, Germany) according to the manufacturer’s instructions. Extracted and purified DNA samples were eluted in 200 µL of PCR-grade water. Fecal DNA samples were shipped to the Parasitology Reference and Research Laboratory of the National Centre for Microbiology (Majadahonda, Spain) in August 2022 for downstream molecular testing.

### 2.6. Molecular Confirmation and Characterization of Giardia duodenalis

Detection of *G. duodenalis* DNA was achieved using a real-time PCR (qPCR) method targeting a 62-bp region of the gene, codifying the small subunit ribosomal RNA (*ssu* rRNA) of the parasite [52].

For assessing the molecular diversity of the parasite, we adopted a sequence-based multilocus genotyping (MLST) scheme targeting the genes encoding for the *ssu* rRNA, the glutamate dehydrogenase (*gdh*), β-giardin (*bg*), and triose phosphate isomerase (*tpi*) proteins of the parasite. For assessing the molecular diversity of *G. duodenalis* at the assemblage level, a nested PCR was used to amplify a 300-bp fragment of the *ssu* rRNA gene [53,54]. The molecular diversity of the parasite at the sub-assemblage level was investigated only in *Giardia* isolates that tested positive by qPCR and yielded cycle threshold (C_T_) values ≤ 32. A semi-nested PCR was used to amplify a 432-bp fragment of the *gdh* gene [55], and nested PCRs were used to amplify 511 and 530 bp fragments of the *bg* and *tpi* genes, respectively [56,57].

### 2.7. General PCR and Electrophoretic Procedures

Detailed information on the PCR cycling conditions and oligonucleotides used for the molecular identification and/or characterization of the protozoan parasites investigated in the present study is presented in Appendix A, respectively. All the direct, semi-nested, and nested PCR protocols described above were conducted on a 2720 Thermal Cycler (Applied Biosystems Foster City, CA, USA). Reaction mixes always included 2.5 units of MyTAQ^TM^ DNA polymerase (Bioline GmbH, Luckenwalde, Germany), and 5–10 µL MyTAQ^TM^ Reaction Buffer containing 5 mM dNTPs and 15 mM MgCl_2_. Laboratory-confirmed positive and negative DNA samples of human origin were routinely used as controls and included in each round of PCR. PCR amplicons were visualized on 1.5% D5 agarose gels (Conda, Madrid, Spain) stained with Pronasafe (Conda) nucleic acid staining solutions.

### 2.8. Sanger Sequencing Analyses

Positive PCR products of the expected size were directly sequenced in both directions using appropriate internal primer sets (Appendix A). DNA sequencing was conducted by capillary electrophoresis using the BigDye^®^ Terminator chemistry (Applied Biosystems) on an ABI PRISM 3130 automated DNA sequencer. Generated DNA consensus sequences were aligned to appropriate reference sequences using MEGA X [58] for species confirmation and genotype identification. The *G. duodenalis* sequences obtained in this study have been deposited in GenBank under accession numbers OP946920–OP946928 (*ssu* rRNA locus), OP947099–OP947117 (*gdh* locus), OP947118–OP947130 (*bg* locus), and OP947131–OP947138 (*tpi* locus).

### 2.9. Haplotype Variability

Significant variables for haplotype variability analysis were selected applying Categorical Principal Components Analysis (CATPCA) and conducting the TwoStep Cluster Analysis procedure through IBM SPSS Statistics for Windows, Version 28.0 (IBM Corp., Armonk, NY, USA) [59], with the purpose of revealing natural groupings within the dataset of continuous and categorical variables and their most relevant input predictors. Cronbach’s Alpha overall value was considered to measure the internal dataset’s consistency.

For the TwoStep Cluster Analysis, the number of clusters was not provided in advance, and Bayesian Information Criterion (BIC) was used to determine the best cluster solution. The input predictor with the highest values of importance was then considered for the haplotype variability analysis and network.

## 3. Results

In this study, 311 stool samples and corresponding questionnaires were collected from asymptomatic school children. Of these, 145 (46.6%) were male and 165 (53.1%) were female. Gender was unknown for a single (0.3%) child. The age of the participating school children ranged from five to 17 years with a median age of 10 years. This rate did not differ between *Giardia*-infected and uninfected school children.

### 3.1. Microscopy

Overall, 31.5% (98/311, 95% Confidence Interval (CI): 26.4–36.7) of fecal samples examined by microscopy tested positive for at least one intestinal parasite (Table 2*). Giardia duodenalis* was the most prevalent parasite found (29.3%, 91/311; 95% CI: 24.3–34.7), followed by *Entamoeba* spp. (18.7%, 58/311; 14.5–23.4), *Ascaris lumbricoides* (1.3%, 4/311; 0.4–3.3), and *Taenia* sp. (0.3%, 1/311; 0.01–1.8). *Giardia duodenalis* was more frequently found in coinfection with *Entamoeba* spp. (15.8%, 49/311) than in monoinfection (11.9%, 37/311). Out of the 91 *Giardia*-positive samples, 18.7% (17/91) corresponded to smears yielding discrepant results at microscopy examination (Appendix A).

### 3.2. Confirmation of G. duodenalis by qPCR

All 91 fecal DNA samples yielding positive/discrepant results for *G. duodenalis* at microscopy examination were subjected to qPCR testing to confirm the presence of the parasite. qPCR positive results were obtained in 76.9% (70/91) of the samples, including 58.8% (10/17) of those generating discrepant results at microscopy. Yielded C_T_ values ranged from 20.0 to 39.6 (median: 30.0; standard deviation: 5.1). Most (65.7%, 46/70) of the qPCR-positive samples yielded C_T_ values ≤ 32.

### 3.3. Genotyping and Subgenotyping of G. duodenalis Isolates

All 91 fecal DNA samples yielding positive/discrepant results for *G. duodenalis* at microscopy examination were subjected to nested *ssu*-PCR to ascertain the assemblage of the parasite involved. Of them, 63.7% (58/91) were successfully genotyped at this locus. These included two isolates that tested negative by qPCR (Appendix A). Sequence analyses revealed the presence of assemblage A (29.3%, 17/58) and assemblage B (70.7, 41/58). Appendix A summarizes the molecular data generated at the *ssu* rRNA locus. Out of the 17 assemblage A sequences, 16 showed 100% identity with the reference sequence used (GenBank accession number AF199446), with the remaining one differing from it by a single nucleotide polymorphism (SNP). Out of the 41 assemblage B sequences, 85.4% (35/41) were identical to reference sequence AF199447, with the remaining six sequences differing by 1–2 SNPs from it. All SNPs detected corresponded to ambiguous (double peak) positions (Appendix A).

The molecular diversity of *G. duodenalis* at the sub-assemblage level was investigated at three (*gdh*, *bg*, and *tpi*) genetic markers. All 46 *Giardia*-positive DNA samples yielding qPCR C_T_ values ≤ 32 were re-assessed under this scheme. Successful PCR amplifications and sequencing data were generated for 56.5% (26/46), 50.0% (23/46), and 26.1% (12/46) of the samples investigated at the *gdh*, *bg*, and *tpi* loci, respectively.

Overall, 65.9% (60/91) of the *Giardia*-positive samples were successfully genotyped at one locus at least (Table 3). MLST data at the four assessed loci were available for 11.7% (7/60) of samples. Subtyping data at a single locus, two loci, or three loci were available for 48.3% (29/60), 18.3% (11/60), and 21.7% (13/60) of samples, respectively. Assemblage B (68.3%, 41/60) was more prevalent than assemblage A (28.3%, 17/60). Mixed A + B infections were detected in two samples (3.3%, 2/60). No host-adapted assemblages of canine (C, D), feline (F), or livestock (E) origin were detected.

Appendix A shows the frequency and molecular diversity of *G. duodenalis* at the *gdh* locus. Out of the 26 *gdh* sequences, nine (34.6%) were assigned to the sub-assemblage AII. Of them, eight showed 100% identity with reference sequence L40510. Assemblages BIII and BIV were identified in 10 (38.5%) and two (7.7%) isolates, respectively. All 10 BIII sequences were different among them, differing by 2–8 SNPs with reference sequence AF069059. The same degree of high genetic variability was observed in the two BIV sequences, which differed by 1–4 SNPs with reference sequence L40508. Discordant BIII/BIV results were identified in an additional five sequences showing 8–13 SNPs when compared with reference sequence L40508.

Appendix A shows the frequency and molecular diversity of *G. duodenalis* at the *bg* locus. Out of the 23 *bg* sequences, eight were identified as sub-assemblage AII. Seven of them were identical to reference sequence AY072723. Four additional sequences were assigned to sub-assemblage AIII, all of them showing 100% identity with reference sequence AY072724. A total of 11 sequences were identified as assemblage B (the *bg* locus is not suitable for sub-assemblage discrimination), of which two were identical to reference sequence AY072727. The remaining nine sequences differed from it by 1–13 SNPs.

Appendix A shows the frequency and molecular diversity of *G. duodenalis* at the *tpi* locus. Out of the 12 *tpi* sequences, five were identified as sub-assemblage AII, all of them identical to reference sequence U57897. Six sequences were unambiguously characterized as sub-assemblage BIII, differing by 1–4 SNPs with reference sequence AF069561. An additional sequence yielded a discordant BIII/BIV result, differing by five SNPs with reference sequence AF069560. As in the case of *gdh*, most of these SNPs corresponded to double peaks.

### 3.4. Haplotype Variability

CATPCA Cronbach’s Alpha overall value (alpha = 0.99) showed evidence of good dataset reliability. The TwoStep Cluster Analysis revealed the existence of two clusters with a silhouette measure of cohesion and separation of 0.3 (fair). The first one represented 43.7% of individuals and the second one included 56.3% of children, with a relationship between the two of 1.29. The variable *Sampled area* resulted to be the input predictor with the highest value (Appendix A); for this reason, it was selected as the main variable for the haplotype variability analysis.

Twenty-six different haplotypes were identified at the *gdh* locus, with the highest haplotype diversity (Hd) value of 0.9917 detected in the *tpi* locus. Haplotypes variability results are displayed in detail in Table 4 and visualized as network representations in Figure 1.

The haplotypes analysis shows assemblage B as having the highest variability, with few mutation steps between the several haplotypes detected, in contrast with assemblage A’s low diversity levels.

For all three loci considered, the haplotypes presenting the highest frequency of detection were not specific for a given area; indeed, they were near equally distributed between at least two different sampling areas, excepting assemblage A at the *tpi* locus (Figure 1C). There is no evidence of a specific haplotype segregation pattern related to a single geographical area, to contiguous sampling areas or to a specific molecular marker.

## 4. Discussion

In this microscopy-based study, one out of three apparently healthy school children were infected with intestinal parasites in Ibadan, Nigeria. *Giardia duodenalis* was the most frequent pathogen detected (29.3%), being present (alone or in combination) in 91 out of the 97 cases of the subclinical parasitic infections observed. *Giardia duodenalis* was more frequently found in association with members of the *Entamoeba* complex (*n* = 49) than in monoinfection (*n* = 37). Infections with nematode (*A. lumbricoides*) and cestode (*Taenia* sp.) parasites were sporadic (less than five cases).

The microscopy-based *G. duodenalis* infection rate detected (29.3%) was in line with those (3–60%) previously reported in Nigerian asymptomatic children [24,25,26,27]. Other studies conducted in the country have documented infection rates of 0.5–26% in children with diarrhea [29,30,31,32], of 0.6–25% in HIV+ patients [33,34,35], and of 0.6–48% in hospital outpatients [36,37,38,39]. These data were obtained in surveys conducted using microscopy (mostly on single stool samples from participating individuals), a diagnostic method known to lack diagnostic sensitivity [60]. These facts strongly suggest that current microscopy-based figures likely underestimate the true burden of *G. duodenalis* infection in Nigeria, particularly in apparently healthy individuals. Indeed, asymptomatic carriers are known to inadvertently spread the infection either in endemic [61] or non-endemic [62] areas. Under this scenario, highly sensitive PCR-based methods coupled with Sanger sequencing can be useful to (i) improve the detection of asymptomatic cases, and by (ii) determining genetic variants and assessing sources of infection, transmission pathways, and zoonotic potential.

The molecular characterization of microscopy-positive *G. duodenalis* isolates at the assemblage and sub-assemblage levels is the most important contribution of this study. This is particularly relevant in Nigeria, a country where molecular-based surveys are scarce. Indeed, human genotyping data is only available for five hospital outpatients in Kaduna state, all of them infected by the assemblage A of the parasite [37]. Zoonotic assemblages A and B have also been reported in goats and pigs [46,47] and in farmed rabbits [48]. Our study is also the first investigation adopting an MLST scheme to assess the genetic diversity of *G. duodenalis* in isolates of human origin, as the only previous survey available used the *tpi* locus only [37]. Of note, MLST analyses are relevant for investigating the epidemiology of *G. duodenalis* because they assist in detecting mixed infections involving different assemblages/sub-assemblages of the parasite and in characterizing sources of infection and zoonotic potential [63].

In our hands, the *ssu*-PCR yielded a higher PCR amplification success rate (63%) than the PCR protocols targeting the *gdh*, *bg*, and *tpi* loci (26–57%), even considering that the latter were only attempted in fecal DNA samples with qPCR C_T_ values ≤ 32. This discrepancy can be explained by the superior diagnostic sensitivity of the multi-copy *ssu* rRNA gene compared to the single-copy *gdh*, *bg*, and *tpi* genes. Our typing analyses revealed that assemblage B was more prevalent than assemblage A (72% vs. 28%) in the pediatric population surveyed. This finding is in line with those previously reported in molecular surveys conducted in human populations in other sub-Saharan African countries including Angola (64% vs. 36%) [64], Ethiopia (82% vs. 18%) [65], Mozambique (90% vs. 8%) [21], Rwanda (86% vs. 13%) [9], Tanzania (79% vs. 21%) [66], and Zambia (73% vs. 27%) [67], among others. Remarkably, no canine-adapted (C, D), feline-adapted (F), or livestock-adapted (F) assemblages were found, suggesting that transmission of giardiasis in the surveyed pediatric population is primarily of anthroponotic nature. However, it should be noted that both assemblages A and B have zoonotic potential [68], so we cannot rule out the possibility that an unknown fraction of the *G. duodenalis* infections detected here were indeed of non-human origin. Additionally, previous molecular-based epidemiological studies conducted in Brazil and Syria have evidenced that sub-assemblage AII was more prevalently found in young children than in individuals of older ages [69,70]. In the present study, AII results were found in 20% (12/60) of the participating children. It would be interesting to see whether this proportion remains unchanged in further studies targeting individuals of all age groups.

In line with the results generated in previous molecular studies in endemic areas [21,71,72,73,74], a much higher molecular variability was observed in assemblage B sequences than in assemblage A sequences regardless the genetic marker used and the geographical origin of the samples. A significant proportion of B sequences (5/17 within the *gdh* marker and 1/12 within the *tpi* marker) corresponded to inconsistent BIII/BIV results. Most of the ambiguous nucleotidic positions in discordant BIII/BIV sequences involved double peaks, indicative of either true intra-assemblage BIII + BIV mixed infections, or the direct result of a genetic recombination mechanism [75,76]. Of note, prior analyses of sequenced *G. duodenalis* assemblage A [63,77] and assemblage B [78] genomes have demonstrated low phylogenetic resolution at the *gdh*, *bg*, and *tpi* loci. These results evidenced the limitations of current MLST schemes and highlighted the necessity of identifying new markers for accurate and robust molecular typing.

The main strengths of this study include a relatively elevated sample size from an endemic area where little epidemiological information was previously available, and the adoption of a classical MLST scheme to assess the molecular diversity within *G. duodenalis*. However, some limitations might have biased the results obtained and the conclusions reached. First, *G. duodenalis* typing and sub-typing analyses were conducted on microscopy-positive samples only. It is likely that performing PCR and Sanger sequencing in all samples would have yielded different frequencies and diversities of *G. duodenalis* assemblages/sub-assemblages. Second, our MLST approach was based on single-copy markers. Adopting more sensitive MLST schemes can results on the assignation of different genetic variants. Third, interviewed children might have provided inaccurate answers to some of the questions included in the epidemiological questionnaire, potentially biasing the results of our statistical analyses. Fourth, epidemiological and molecular data generated in the present study is restricted to the age group and geographical area investigated, and might not be representative of the entire population in Nigeria. And fifth, initial detection of intestinal parasites was based on microscopy examination of single stool samples. Because of the limited diagnostic sensitivity of this method and the irregular shedding of the parasite’s transmission stages (eggs, cysts, oocysts, spores) in stools from infected patients, reported prevalence rates might represent an underestimation of the true ones.

## 5. Conclusions

Our results indicate that *G. duodenalis* is a common finding in apparently healthy school children in Ibadan, Nigeria. This study provides the most comprehensive and thorough account on the molecular diversity of *G. duodenalis* in this country. As in other sub-Saharan African regions, assemblage B is the predominant genetic variant of the parasite circulating in humans. This fact, together with the absence of animal-adapted assemblages in the surveyed pediatric population, strongly suggests that human transmission of *G. duodenalis* is primarily anthroponotic. Therefore, efforts to control giardiasis (and other fecal-orally transmitted pathogens) should focus on providing safe drinking water and improving sanitation and personal hygiene practices including hand washing.

## Figures and Tables

**Figure 1 tropicalmed-08-00152-f001:**
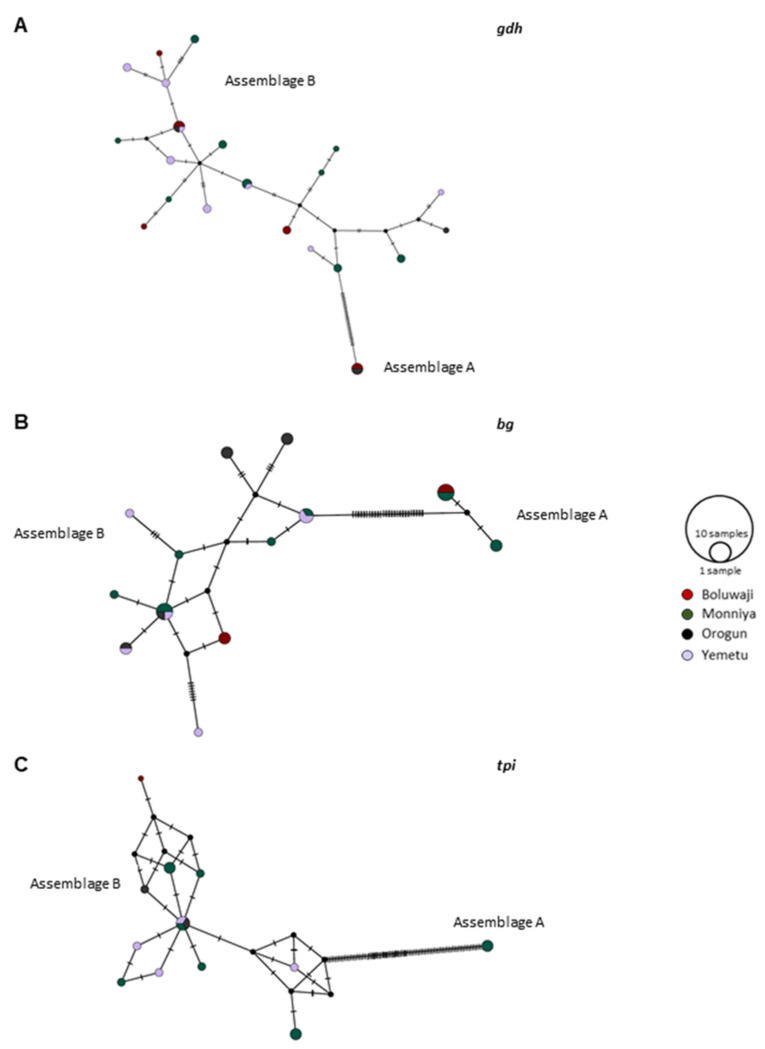
Median-joining haplotype networks built in PoPArt software using *gdh* (**A**), *bg* (**B**), and *tpi* (**C**) sequences generated in the present study. Haplotypes are represented by circles proportional to relative haplotypes abundance; different colors indicate different sampling areas. Hatch marks refer to the mutational steps between haplotypes. Black circles represent hypothetical missing haplotypes predicted.

**Table 1 tropicalmed-08-00152-t001:** Occurrence and genetic diversity of *Giardia duodenalis* in human and animal hosts in Nigeria, 1994–2022.

Host Population	City/Region	Detection Method	Samples Analyzed (*n*)	Infection Rate (%)	Assemblage (*n*)	Reference
**Human**						
Asymptomatic children	Anambra State	CM	1536	0.1	ND	[23]
Asymptomatic children	Benin City	CM	184	59.8	ND	[24]
Asymptomatic children	Imo State	CM	300	2.7	ND	[25]
Asymptomatic children	Lagos City	CM	384	12.2	ND	[26]
Asymptomatic children	Niger Delta	CM	1062	9.2	ND	[27]
Asymptomatic children	Oyo State	PCR, DFA	199	37.2	ND	[28]
Diarrheic children	Ebonyi State	PCR, DFA	199	0.0	–	[29]
Diarrheic children	Ilesa City	CM	300	7.1	ND	[30]
Diarrheic children	Lagos City	CM	215	0.5	ND	[31]
Diarrheic children	Zaria City	CM	142	26.1	ND	[32]
HIV+ patients	Benin City	CM	2000	0.6	ND	[33]
HIV+ patients	Lagos City	CM	65	9.2	ND	[34]
HIV+ patients	Zaria City	CM	10	25.0	ND	[35]
Hospital outpatients	Enugu State	CM	500	47.8	ND	[36]
Hospital outpatients	Ibadan City	CM	360,000	1.6	ND	[37]
Hospital outpatients	Kaduna State	PCR-RFLP	157	3.2	A (5)	[38]
Hospital outpatients	Lagos City	CM	1015	0.6	ND	[34]
Hospital outpatients	Ogun State	CM	479	4.2	ND	[39]
Food handlers	Ogun State	CM	100	13.0	ND	[40]
Game scouts	Kainji Lake National Park	CM	20	NS	ND	[41]
Rural dwellers	Plateau State	CM	300	2.3	ND	[42]
Urban dwellers	Enugu State	CM	403	35.2	ND	[43]
Urban dwellers	Jos City	CM	111	4.3–7.2	ND	[44]
Urban dwellers	Lagos City	CM	2099	7.9	ND	[45]
**Livestock**						
Goat	Ogun State	ELISA, PCR	302	45.7	A (5), B (13), E (40)	[46]
Pig	Ogun State	ELISA, PCR	209	25.4	B (4), E (37), B + E (2)	[47]
Rabbit	Ogun State	PCR	83	72.3	B (19)	[48]
**Insects**						
Cockroach	Lagos State	CM	749	18.7	ND	[49]
Fly	Ogun State	CM	7190	3.3	ND	[50]

CM: conventional microscopy; DFA: direct fluorescence assay; ELISA: enzyme-linked immunosorbent assay; ND: not determined; NS: not specified; PCR: polymerase chain reaction; PCR-RFLP: polymerase chain reaction-restriction fragment length polymorphism.

**Table 2 tropicalmed-08-00152-t002:** Detection of intestinal parasites by microscopy examination.

Parasite Species	Positive (*n*)	Frequency (%) ^1^
None	213	68.5
**In monoinfection**		
*G. duodenalis*	37	11.9
*Entamoeba* complex	7	2.3
**In coinfection**		
*G. duodenalis* + *A. lumbricoides*	2	0.6
*G. duodenalis* + *Entamoeba* complex	49	15.8
*G. duodenalis* + *Taenia* sp.	1	0.3
*G. duodenalis* + *A. lumbricoides* + *Entamoeba* complex	2	0.6
Total	311	100

^1^ Over the total of fecal samples (*n* = 311) examined. *Entamoeba* complex: *Entamoeba histolytica*/*dispar*/*moshkovskii*.

**Table 3 tropicalmed-08-00152-t003:** Multilocus sequence typing results of the 60 *G. duodenalis*-positive samples of pediatric origin successfully genotyped at any of the four loci investigated in the present survey. The age and gender of the infected children are also shown.

Sample ID	Age (Years)	Gender	C_T_ Value in qPCR	*ssu* rRNA	*gdh*	*bg*	*tpi*	Assigned Genotype
2	8	M	25.5	–	–	B	–	B
4	9	F	29.4	B	–	–	–	B
5	12	F	27.8	B	BIII	–	–	BIII
6	11	F	–	B	–	–	–	B
9	10	M	23.6	B	BIII	–	–	BIII
11	9	F	32.6	B	–	–	–	B
13	12	M	27.9	B	BIII	–	–	BIII
15	7	M	30.3	–	BIII	–	–	BIII
17	11	F	25.8	B	–	–	–	B
25	6	M	34.8	B	–	–	–	B
27	5	F	31.5	B	–	–	–	B
28	7	M	31.5	B	–	–	–	B
29	8	F	32.7	B	–	–	–	B
30	8	M	38.6	B	–	–	–	B
31	7	M	21.8	B	BIII/BIV	B	–	BIII/BIV
32	10	M	–	A	–	–	–	A
33	9	M	21.2	A	AII	AII	–	AII
34	12	F	30.6	B	BIII	–	–	BIII
36	10	M	23.6	B	–	B	–	B
37	11	F	25.7	A	–	–	–	A
39	10	F	22.2	A	AII	AII	–	AII
41	10	M	20.0	B	–	–	–	B
44	7	M	24.9	B	–	AII	–	AII + B
46	5	F	31.2	B	BIII	–	–	BIII
47	7	F	24.7	B	BIII	B	–	BIII
49	15	M	29.8	B	–	–	–	B
50	8	M	24.7	A	AII	AII	–	AII
51	5	M	27.8	B	BIII	–	–	BIII
52	9	F	21.2	A	AII	AII	AII	AII
53	10	F	23.7	A	AII	AIII	–	AII/AIII
54	10	F	23.1	A	AII	AIII	–	AII/AIII
55	8	F	23.1	A	AII	AIII	–	AII/AIII
56	14	F	35.8	A	–	–	–	A
58	11	F	36.3	A	–	–	–	A
59	12	F	24.4	B	BIII/BIV	B	BIII	BIII/BIV
61	10	F	24.1	B	BIII/BIV	B	–	BIII/BIV
63	12	F	31.4	B	BIV	–	–	BIV
68	13	M	26.8	B	BIII	B	BIII	BIII
70	11	M	27.3	A	–	AII	AII	AII
77	11	F	34.3	A	–	–	–	A
78	11	M	24.9	A	AII	AII	AII	AII
79	12	M	33.6	B	–	–	–	B
80	14	F	34.6	B	–	–	–	B
81	5	M	33.3	A	–	–	–	A
82	10	M	22.3	B	–	–	BIII	BIII
83	13	F	31.0	B	–	–	–	B
84	9	M	26.7	B	BIII/BIV	B	BIII	BIII/BIV
85	14	F	23.2	B	–	B	BIII	BIII
88	6	F	22.0	B	BIV	–	BIII/BIV	BIII/BIV
89	10	M	23.0	B	BIII/BIV	B	BIII	BIII/BIV
91	8	F	22.5	A	AII	AII	AII	AII
92	7	F	32.6	B	–	B	–	B
94	7	F	36.3	B	–	–	–	B
95	9	F	28.3	B	–	–	–	B
96	11	F	29.9	B	–	–	–	B
97	10	F	32.5	B	BIII	–	AII	AII + BIII
98	10	M	30.6	B	–	–	–	B
99	3	M	30.2	B	–	–	–	B
100	10	M	29.1	B	–	–	–	B
102	9	F	24.7	A	–	AIII	–	AIII

**Table 4 tropicalmed-08-00152-t004:** Haplotypes distribution for *gdh*, *bg,* and *tpi* sequences.

Marker	Sequences (*n*)	Sites (*n*)	Monomorphic Sites (*n*)	Segregating Sites (*n*)	Segregating Sites with >2 Variants (*n*)	Positions with Gaps (*n*)	Positions with Missing Data (*n*)	Variable Sites (*n*)	Haplotypes (*n*)	Haplotype Diversity
*gdh*	38	433	238	63	15	132	0	180	26	0.9815
*bg*	26	637	388	53	10	196	0	240	19	0.9785
*tpi*	16	531	299	108	27	124	0	208	15	0.9917

## Data Availability

All relevant data are within the article and its Appendix A. The sequences data were submitted to the GenBank database under the accession numbers OP946920–OP946928 (*ssu* rRNA locus), OP947099–OP947117 (*gdh* locus), OP947118–OP947130 (*bg* locus), and OP947131–OP947138 (*tpi* locus).

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
