# Peer review of "High Diversity of Giardia duodenalis Assemblages and Sub-Assemblages in Asymptomatic School Children in Ibadan, Nigeria"

_tropicalmed, 2023, doi:10.3390/tropicalmed8030152_

Round 1

Reviewer 1 Report

ABSTRACT

PAGE 1, LINE 27: Please revise the species or group of species; see also this issue bellow at the results section

…. followed by Entamoeba spp. (18.7%, 58/311; 14.5–23.4), 27 Ascaris (1.3%, 4/311; 0.4–3.3), and a member of the family Taeniidae….

INTRODUCTION

PAGE 2, LINE 51: “….settings where there is no or inadequate access to clean…..” I suggest changing to “…. settings without or with inadequate access to clean….”

PAGE 2, LINE 69: “Although Giardia infection is not usually associated with mortality, giardiasis seems widely present in the country, affecting both asymptomatic and clinical individuals and livestock (Table 1)”. I suppose it is worth to mention here that despite the different studies in humans, little is known about the Giardia assemblages in Nigeria (Table 1 shows only one study with five A assemblages characterized).

MATERIALS AND METHODS

PAGE 3, LINE 80: 2.1. Study design and participants

Number of participants is missing in this subsection

PAGE 3, LINE 96: 2.3. Epidemiological questionnaires

“Individual standardized questionnaires in Yoruba and English languages were completed by a member of our research team in face-to-face interviews with each participating child at sample collection.” Considering young children, questionnaires should be answered by their parents or legal guardians.

PAGE 4 LINE 117 - Giardia duodenalis; correct italics

RESULTS

PAGE 5, LINE 177:

Giardia duodenalis was the most…, followed by Entamoeba spp. (18.7%, 58/311; 14.5–23.4), Ascaris (1.3%, 4/311; 0.4–3.3), and a member of the family Taeniidae (0.3%, 1/311; 0.01–1.8).

a)      Except for the Entamoeba histolytica/dispar/moshkovskii complex, that are morphologically identical, other intestinal ameba of the Entamoeba genus, such E. coli and E. hartmanni can be microscopically distinguished by their size/morphology. I suggest to the authors clarify which is the group or species of Entamoeba in table 2.

b)      The species Ascaris lumbricoides is the most common roundworm in humans

c)      I also suggest writing  Taenia sp instead of member of the family Taeniidae

PAGE 5, LINE 193:

3.3. Genotyping and subgenotyping of G. duodenalis isolates

In this subsection I think that the supplemental tables information is too long. I suggest summarizing this topic and perhaps to combine some Tables, such S5, S6 and S7 that describe the frequency and molecular diversity of G. duodenalis at the different loci.

DISCUSSION

PAGE 9, LINE 273:

“Remarkably, G. duodenalis was more frequently found in association with members of the Entamoeba complex.” These parasites are intestinal protozoa transmitted mostly by the fecal-oral route, with contaminated drinking water being a very important vehicle for cysts transmission. Besides, they can be diagnosed by the same parasitological methods. In poor settings is very common co-infections. Please, delete “remarkably”

PAGE 10, LINE 278:

“Other studies conducted in the country have documented infection rates of 0.5–26% in children with diarrhea, of 0.6–25% in HIV+ patients, and of 0.6–48% in hospital outpatients (Table 1).” Add the specific references for these studies, instead referring to Table 1.

PAGE 10, LINE 281:

“These data were obtained in surveys conducted using microscopy, a diagnostic method known to lack diagnostic sensitivity”. The use of Giardia coproantigen detection by ELISA or rapid tests would probably enhance the number of cases diagnosed, including in this study. The PCR-based methods are essential to characterize the genetic variants of Giardia, but have their own limitations when used for diagnosis. Indeed, the highest PCR amplification success rate in this work was at the ssu rRNA (63%). Please see the reference bellow:

Silva RK, Pacheco FT, Martins AS, et al. Performance of microscopy and ELISA for diagnosing Giardia duodenalis infection in different pediatric groups. Parasitol Int. 2016;65(6 Pt A):635-640. doi:10.1016/j.parint.2016.08.012

PAGE 10, LINE 314:

“However, it should be noted that both assemblages A and B have zoonotic potential [68], so we cannot rule out the possibility that an unknown fraction of the G. duodenalis infections detected here were indeed of non-human origin.”

Usually, the finding of AII sub-genotype suggests that transmission of giardiasis occurs mainly through an anthroponotic route (direct or indirect) since this subtype is predominantly isolated from humans. Besides, it seems to occur more frequently in young children. Authors could discuss briefly this issue, as they found some AII genetic variants. See the references:

Pacheco FTF, Silva RKNR, Carvalho SS, et al. The Predominance of Giardia duodenalis AII sub-assemblage in young children from Salvador, Bahia, Brazil. Biomedica. 2020 Sep 1;40(3):557-568. doi: 10.7705/biomedica.

Skhal D, Aboualchamat G, Al Mariri A, Al Nahhas S. Prevalence of Giardia duodenalis assemblages and sub-assemblages in symptomatic patients from Damascus city and its suburbs. Infect Genet Evol. 2017;47:155-160. doi:10.1016/j.meegid.2016.11.030.

Author Response

Reviewer #1

  1. PAGE 1, LINE 27: Please revise the species or group of species; see also this issue bellow at the results section…. followed by Entamoeba (18.7%, 58/311; 14.5–23.4), 27 Ascaris (1.3%, 4/311; 0.4–3.3), and a member of the family Taeniidae….

Reply: displayed information is correct.

  1. PAGE 2, LINE 51: “….settings where there is no or inadequate access to clean…..” I suggest changing to “…. settings without or with inadequate access to clean….”

Reply: modified as per requested.

  1. PAGE 2, LINE 69: “Although Giardia infection is not usually associated with mortality, giardiasis seems widely present in the country, affecting both asymptomatic and clinical individuals and livestock (Table 1)”. I suppose it is worth to mention here that despite the different studies in humans, little is known about the Giardia assemblages in Nigeria (Table 1 shows only one study with five A assemblages characterized).

Reply: we agree with Reviewer #1. The sentence “In addition, very little is currently known on the genetic variability of G. duodenalis circulating in Nigerian human populations.” has been now added into current lines 74-76 of the Introduction section.

  1. PAGE 3, LINE 80: 2.1. Study design and participants. Number of participants is missing in this subsection

Reply: please note that this information is provided in the first line of the Results section in current line 174.

  1. PAGE 3, LINE 96: 2.3. Epidemiological questionnaires. “Individual standardized questionnaires in Yoruba and English languages were completed by a member of our research team in face-to-face interviews with each participating child at sample collection.” Considering young children, questionnaires should be answered by their parents or legal guardians.

Reply: please note that questionnaires were completed by interviewing children at school at the moment of sample collection because i) some parents/legal guardians  were illiterate and were not able to fill the questionnaires at home, and ii) for practical and logistic reasons it was impossible to interview parents and children at the same time during sampling collection. We agree that interviewed children might have provided inaccurate answers to some of the questions included, potentially biasing the results of the statistical analyses. We have acknowledged this possibility in the paragraph devoted to the limitations of the study in current lines 349-351 of the Discussion section.

  1. PAGE 4 LINE 117 - Giardia duodenalis; correct italics

Reply: corrected as per requested.

  1. PAGE 5, LINE 177: Giardia duodenalis was the most…, followed by Entamoeba (18.7%, 58/311; 14.5–23.4), Ascaris (1.3%, 4/311; 0.4–3.3), and a member of the family Taeniidae (0.3%, 1/311; 0.01–1.8).

Reply: as commented in our answer to your comment #1, displayed information is correct.

  1. Except for the Entamoeba histolytica/dispar/moshkovskii complex, that are morphologically identical, other intestinal ameba of the Entamoeba genus, such coli and E. hartmanni can be microscopically distinguished by their size/morphology. I suggest to the authors clarify which is the group or species of Entamoeba in table 2.

Reply: clarified as per requested in current Table 2.

  1. The species Ascaris lumbricoides is the most common roundworm in humans

Reply: Following Reviewer #1 recommendation, the term “Ascaris” has been replaced by “Ascaris lumbricoides” or “A. lumbricoides” where appropriate through the main body of the text and Table 2.

  1. I also suggest writing Taenia sp instead of member of the family Taeniidae

Reply: Following Reviewer #1 recommendation, the term “member of the family Taeniidae” has been replaced by “Taenia sp.” where appropriate through the main body of the text and Table 2.

  1. PAGE 5, LINE 193: 3.3. Genotyping and subgenotyping of duodenalis isolates. In this subsection I think that the supplemental tables information is too long. I suggest summarizing this topic and perhaps to combine some Tables, such S5, S6 and S7 that describe the frequency and molecular diversity of G. duodenalis at the different loci.

Reply: We thank Reviewer #1 for his/her comment, which we have seriously considered. However, we feel that information contained in current lines 225-247 are necessary to properly explain the marked differences observed in the genetic variability between assemblage A and assemblage B sequences particularly at the gdh and tpi loci. This information would be lost if we eliminate data on homology with reference sequences or presence of SNPs, particularly taking into account that the Tables where this information can be visualized are supplemental. For these reasons we would like to keep this part of the Results section as it currently stands.

  1. PAGE 9, LINE 273: “Remarkably, duodenalis was more frequently found in association with members of the Entamoeba complex.” These parasites are intestinal protozoa transmitted mostly by the fecal-oral route, with contaminated drinking water being a very important vehicle for cysts transmission. Besides, they can be diagnosed by the same parasitological methods. In poor settings is very common co-infections. Please, delete “remarkably”

Reply: deleted as per requested.

  1. PAGE 10, LINE 278: “Other studies conducted in the country have documented infection rates of 0.5–26% in children with diarrhea, of 0.6–25% in HIV+ patients, and of 0.6–48% in hospital outpatients (Table 1).” Add the specific references for these studies, instead referring to Table 1.

Reply: specific reference numbers added as per requested.

  1. PAGE 10, LINE 281: “These data were obtained in surveys conducted using microscopy, a diagnostic method known to lack diagnostic sensitivity”. The use of Giardia coproantigen detection by ELISA or rapid tests would probably enhance the number of cases diagnosed, including in this study. The PCR-based methods are essential to characterize the genetic variants of Giardia, but have their own limitations when used for diagnosis. Indeed, the highest PCR amplification success rate in this work was at the ssu rRNA (63%). Please see the reference bellow: Silva RK, Pacheco FT, Martins AS, et al. Performance of microscopy and ELISA for diagnosing Giardia duodenalis infection in different pediatric groups. Parasitol Int. 2016;65(6 Pt A):635-640. doi:10.1016/j.parint.2016.08.012

Reply: please note that the superior performance of the ssu-PCR is highly expected because the targeted marker (ssu rRNA) is a multicopy gene with enhanced diagnostic sensitivity compared that those of single-copy genes such as the gdh, bg, and tpi markers used in this study for genotyping and subgenotyping purposes. Please note that this limitation of the study has been already acknowledged in current lines 309-311 and 347.

We agree with Reviewer #1 that by adopting ELISA (instead of conventional microscopy) as screening test we would probably have increased the number of Giardia-positive samples (providing a more accurate prevalence rate) and the number of genotyped isolates. However, we feel that sequencing and genotyping 60 sequences (as we already did) provides a good enough estimation of the frequency and diversity of G. duodenalis assemblages/sub-assemblages. Adding few more positive sequences to this analysis would not modify the final results significantly.

  1. PAGE 10, LINE 314: “However, it should be noted that both assemblages A and B have zoonotic potential [68], so we cannot rule out the possibility that an unknown fraction of the duodenalis infections detected here were indeed of non-human origin.”. Usually, the finding of AII sub-genotype suggests that transmission of giardiasis occurs mainly through an anthroponotic route (direct or indirect) since this subtype is predominantly isolated from humans. Besides, it seems to occur more frequently in young children. Authors could discuss briefly this issue, as they found some AII genetic variants. See the references:

Pacheco FTF, Silva RKNR, Carvalho SS, et al. The Predominance of Giardia duodenalis AII sub-assemblage in young children from Salvador, Bahia, Brazil. Biomedica. 2020 Sep 1;40(3):557-568. doi: 10.7705/biomedica.

Skhal D, Aboualchamat G, Al Mariri A, Al Nahhas S. Prevalence of Giardia duodenalis assemblages and sub-assemblages in symptomatic patients from Damascus city and its suburbs. Infect Genet Evol. 2017;47:155-160. doi:10.1016/j.meegid.2016.11.030.

Reply: Following Reviewer#1 recommendation, we have added the following paragraph “Additionally, previous molecular-based epidemiological studies conducted in Brazil and Syria have evidenced that sub-assemblage AII was more prevalently found in young children than in individuals of older age [69,70]. In the present study, AII results were found in 20% (12/60) of the participating children. It would be interesting to see whether this proportion remains unchanged in further studies targeting individuals of all age groups.” in current lines 322-327 of the Discussion section. The references Figueiredo Pacheco et al. Biomedica 2020;40:557–568 and Skhal et al. Infect Genet Evol 2017;47:155–160 have been added to the reference list and the subsequent references renumbered accordingly.

Reviewer 2 Report

The manuscript is interesting, however, there are several details that must be considered before it can be considered for publication.

Abstract

As it is not structured, it is difficult to understand. Some prayers are superfluous and should not be in this essential section, such as the other parasites found.

Introduction

The second paragraph does not provide relevant information to contextualize the study. Table 1 is confusing and doesn't deserve to be in the job, mainly because it doesn't provide the assemblages.

Methods

There is no description of the ethical aspects (coding, anonymization, etc.)

What was the reason for not directly analyzing all the samples obtained with PCR?

Results.

Supplementary tables are not available.

Discussion

You must consider the zoonotic importance of the other assemblages of the parasite and the reasons for not having found them. I suggest using the following manuscript: https://doi.org/10.54034/mic.e1268

Conclusion

Focus on the research question.

Author Response

Reviewer #2

The manuscript is interesting, however, there are several details that must be considered before it can be considered for publication.

Reply: We thank Reviewer #2 for his/her preliminary appraisal.

  1. As it is not structured, it is difficult to understand. Some prayers are superfluous and should not be in this essential section, such as the other parasites found.

Reply: please note that the editing format of Tropical Medicine and Infectious Diseases specifically requests unstructured abstracts of a maximum of 200 words. This is exactly what we have provided in our manuscript. Regarding the difficulty of understanding the abstract, please note that none of the other three external reviewers nor the Associated Editor in charge of the review process have commented on this matter.

We disagree with Reviewer #2 recommendation of removing prevalence data from other intestinal parasites. Please note that this is a microscopy-based study, so we believe that results from other pathogens complementing those obtained for Giardia are an asset of the manuscript.

  1. The second paragraph does not provide relevant information to contextualize the study. Table 1 is confusing and doesn't deserve to be in the job, mainly because it doesn't provide the assemblages.

Reply: we strongly disagree with Reviewer #2. Please note that, in African endemic areas, G. duodenalis is typically more frequently found in children without diarrhea than in children with diarrhea (see current references Kotloff Lancet 2013, 382, 209–222; Platts-Mills Lancet Glob Health 2015 3, e564-75; Donowitz Clin Infect Dis 2016 63, 792–797). This is precisely why Giardia is not included in the Global Burden of Disease (GBD) studies specifically devoted to diarrhea-causing pathogens. Despite this, Giardia infection is widely known as an agent associated to impaired growth and cognitive development in affected children living in endemic areas such as Nigeria. We believe that explaining these facts is essential to put into context the epidemiological scenario and clinical relevance of giardiasis in sub-Saharan Africa in general and Nigeria in particular. This is exactly what we attempted to do in the second paragraph of the Introduction section.

Regarding Table 1, we again disagree with Reviewer #2. We included the Table to summarize all the information available on the epidemiology of Giardia duodenalis in human and animal hosts and use this information to put into context our data in the Discussion section. We purposely use this Table to highlight the lack of genotyping data on human isolates in the country (on this matter see also our answer to main comment by Reviewer #3), with only a single study providing such information. Please also note that most of the studies included in the Table were conducted by microscopy, a method unable to provide molecular data. This is why assemblage information is missing in these studies (as clearly indicated by the abbreviation ND: not determined). Please see also our answer to comment #4 by Reviewer #1, who acknowledged the usefulness of Table 1.

  1. There is no description of the ethical aspects (coding, anonymization, etc.)

Reply: please note that ethical issues have been disclosed in the appropriate section (Institutional Review Board Statement) in current lines 390-392. A PDF copy of this document (specifying that all procedures involved in sample collection and personal data protection have been approved by the Ethics Committee of the institution of origin) is available under request. In addition, the full dataset of this survey can be found in current Table S3. Reviewer #3 can verify there that there is no link between the data presented and the individual person who provided the sample/data.

  1. What was the reason for not directly analyzing all the samples obtained with PCR?

Reply: the diagnostic algorithm used in this survey used conventional microscopy as screening method and PCR only for the samples that tested positive for Giardia at microscopy examination. We did not analyse the whole sample set because of budget restrictions.

  1. Supplementary tables are not available.

Reply: All supplementary tables were included in the initial submission of the manuscript.

  1. You must consider the zoonotic importance of the other assemblages of the parasite and the reasons for not having found them. I suggest using the following manuscript: https://doi.org/10.54034/mic.e1268

Reply: please note that zoonotic aspects of the results generated in this survey have been already discussed in current lines 299-306 and 319-322. Please also note that in this study we only identified assemble A and assemblage B circulating in the human population under investigation. We already acknowledged that “no canine-adapted (C, D), feline-adapted (F), or livestock-adapted (F) assemblages were found, suggesting that transmission of giardiasis in the surveyed pediatric population is primarily of anthroponotic nature” in current lines 317-319. In addition to this, the paper suggested by Reviewer #2 was conducted in dog, not human, samples, so there is no point in including it in the present study.

  1. Focus on the research question.

Reply: we do not understand what Reviewer 2# means. Actually, we do focus in the research question, which is the description of the molecular frequency and diversity of G. duodenalis in isolates of human origin. Please note that our research group has ample experience on this field. See for instance PMIDs 36282323, 34072858, 32981546, 33668348, 33672794, 33465074, 32218318, 32027684, among many others.

Reviewer 3 Report

The authors present their findings about the occurrence and molecular diversity of G. duodenalis and other intestinal parasites in apparently healthy children (n = 311) in Ibadan, Nigeria. The study lacks novelty since there are several studies published on this topic. Major weakness of this study is that they only performed qPCR in 91 positive samples from microscopy instead of the total collected samples.  

Line 3; 72;; 82;270 add space after school 

Line 574 add other published papers (Tandukar et al., 2018: Gupta et al., 2020) 

Author Response

Reviewer #3

The authors present their findings about the occurrence and molecular diversity of G. duodenalis and other intestinal parasites in apparently healthy children (n = 311) in Ibadan, Nigeria. The study lacks novelty since there are several studies published on this topic. Major weakness of this study is that they only performed qPCR in 91 positive samples from microscopy instead of the total collected samples.

Reply: We thank Reviewer #3 for his/her preliminary appraisal. We would like to highlight that, to the best of our knowledge, only a single molecular-based study has previously investigated the genetic diversity of G. duodenalis in Nigerian human populations (see Table 1 of current manuscript). In that survey, five Giardia-positive patients were found infected by the assemblage A of the parasite as determined at a single (tpi) locus. This is all what we know on the molecular epidemiology of G. duodenalis in humans in Nigeria. Under this scenario, we strongly believe that any additional information on this field would represent a significant contribution to help in bridging this gap of knowledge. Please also note that our survey provides genotyping data for 60 G. duodenalis isolates of human origin using four (ssu rRNA, gdh, bg, and tpi) loci. Few studies in sub-Saharan countries have reported similar findings, so from this point of view we believe that our data are novel and worth of publishing. That said, we agree with Reviewer #3 that data generated in the present study are similar to those from other surveys conducted in sub-Saharan countries, confirming a general pattern (higher frequency of assemblage B over assemblage A, absence of animal-adapted assemblages circulating in humans, predominance of anthropic transmission over zoonotic transmission) in this geographical region of the world.

  1. Line 3; 72; 82;270 add space after school

Reply: corrected as per requested.

  1. Line 574 add other published papers (Tandukar et al., 2018: Gupta et al., 2020)

Reply: please note that in this manuscript the whole Discussion section has focused in the African epidemiological scenario. The above-mentioned references are from studies conducted in Nepal with little or no relationship with our study. Under this premise, we have omitted this request by Reviewer ·#3.

Reviewer 4 Report

This is a well-done molecular study of Giardia assemblies (species) among asymptomatic children in a city in southern Nigeria. Major findings include a high prevalence of Giardia by microscopy of stools, the vast majority of which are types A and B that are human-specific. They make an appropriate inference from the data that hygiene needs to be improved. These results are not new but are an important reminder of the endemic nature of Giardia infections in under-resourced countries.

Author Response

Reviewer #4

This is a well-done molecular study of Giardia assemblies (species) among asymptomatic children in a city in southern Nigeria. Major findings include a high prevalence of Giardia by microscopy of stools, the vast majority of which are types A and B that are human-specific. They make an appropriate inference from the data that hygiene needs to be improved. These results are not new but are an important reminder of the endemic nature of Giardia infections in under-resourced countries.

Reply: We thank Reviewer #4 for his/her preliminary positive appraisal. Regarding his/her comment on the lack of novelty of the results obtained in the present study, please see our answer to main comment by Reviewer #3.

Reviewer 5 Report

Although the study was focused on the detection and characterization of Giardia species and haplotypes I believe that the microscopic study is limited by the use of a single concentration method and a single stain, limiting the number of parasite species detected. The study is well exectuted and presented finding appropiate the content on each section however some information regarding the climatology and hydrology in the study area could have been included in virtue of the influence of climatic factors on the epidemiology of intestinal parasitosis. English language is correct. 

Author Response

Although the study was focused on the detection and characterization of Giardia species and haplotypes I believe that the microscopic study is limited by the use of a single concentration method and a single stain, limiting the number of parasite species detected. The study is well exectuted and presented finding appropiate the content on each section however some information regarding the climatology and hydrology in the study area could have been included in virtue of the influence of climatic factors on the epidemiology of intestinal parasitosis. English language is correct.

Reply: We thank Reviewer #5 for his/her preliminary positive appraisal. We have now acknowledged the limitations of conventional microscopy as screening method in single stool samples in lines 401-404 of the Discussion section. In addition, we have added some climatic data on lines 89-93 of the Material and Methods section.

Reviewer 6 Report

The manuscript is very interesting and important in understanding the molecular epidemiology of Giardia duodenalis.

Authors should mention:

1. In the manuscript it is said:

  "These data were obtained in surveys conducted using microscopy, diagnostic method known to lack diagnostic sensitivity"

Comment: The output of cysts in feces lacks periodicity, so the analysis of three samples collected on different days is recommended.

This requirement is widely recognized and mentioned in documents such as:

a) CLSI M28 2A.

b) Practical guide to diagnostic parasitology / Lynne S. Garcia. 3rd edition. Hoboken, NJ : Wiley, 2021

c) World Health Organization. (‎2019)‎. Bench aids for the diagnosis of intestinal parasites, 2nd ed.. World Health Organization

I suggest should say:

These data were obtained in surveys conducted using microscopy, diagnostic method known to lack diagnostic sensitivity WHEN USING ONLY ONE SAMPLE"

COMMENT: This correction is important, since the coproparasitoscopic method has the necessary sensitivity when the appropriate indications are followed. If the molecular methods used 3 samples from different days they would be much more sensitive, but more expensive in routine work.

2. In the locality under study, it was shown that the main source of Giardiosis infection was from human feces and not from reservoirs. They can comment on how many animals are present in the studied localities, since if there are very few animals, the probability of transmission decreases. It could be the case that despite the existence of abundant animals in the locality, the presence of zoonotic genotypes was very low.

Please can you comment on it.

Author Response

The manuscript is very interesting and important in understanding the molecular epidemiology of Giardia duodenalis.

Reply: We thank Reviewer #6 for his/her preliminary positive appraisal.

  1. In the manuscript it is said:   "These data were obtained in surveys conducted using microscopy, diagnostic method known to lack diagnostic sensitivity". Comment: The output of cysts in feces lacks periodicity, so the analysis of three samples collected on different days is recommended. This requirement is widely recognized and mentioned in documents such as: a) CLSI M28 2A; b) Practical guide to diagnostic parasitology / Lynne S. Garcia. 3rd edition. Hoboken, NJ : Wiley, 2021, c) World Health Organization. (‎2019)‎. Bench aids for the diagnosis of intestinal parasites, 2nd ed.. World Health Organization

I suggest should say: in surveys conducted using microscopy, diagnostic method known to lack diagnostic sensitivity WHEN USING ONLY ONE SAMPLE". COMMENT: This correction is important, since the coproparasitoscopic method has the necessary sensitivity when the appropriate indications are followed. If the molecular methods used 3 samples from different days they would be much more sensitive, but more expensive in routine work.

Reply: following Reviewer #6 advice, the sentence has been rephrased as “These data were obtained in surveys conducted using microscopy (mostly on single stool samples from participating individuals), a diagnostic method known to lack di-agnostic sensitivity” in current lines 290-292.

  1. In the locality under study, it was shown that the main source of Giardiosis infection was from human feces and not from reservoirs. They can comment on how many animals are present in the studied localities, since if there are very few animals, the probability of transmission decreases. It could be the case that despite the existence of abundant animals in the locality, the presence of zoonotic genotypes was very low.

Reply: please note that the statement that our human giardiasis cases are mainly of anthropic nature is based on two findings including i) assemblages A and B were the only G. duodenalis genetic variants circulating in the surveyed asymptomatic schoolchildren population, and ii) canine- (C, D), ungulate- (E), feline- (F) and murine-adapted (G) assemblages were not found in the surveyed children. Although both assemblages A and B are regarded as zoonotic, when considered together, these data pointed out to the predominance of an anthropic origin of those infections.

We also know that 75% (235/311) of participating children (see Table S3) declared regular contact with domestic animals including livestock, dogs, and poultry. Livestock are primarily infected with G. duodenalis assemblage E, dogs with assemblages C/D, and poultry by avian-specific Giardia species including G. ardeae and G. psittaci. None of these genetic variants were identified in the paediatric population under investigation. In the absence of molecular data from animal and (animal-contaminated) environmental samples, we feel that assessing the occurrence and extent of potential zoonotic events would be too hypothetical. This is the reason why we prefer adopting a conservative approach on this matter.

Round 2

Reviewer 2 Report

Dear authors.

The sole purpose of making the suggestions was to improve the content and focus of your manuscript.

Unfortunately, like every other author including myself, we consider our work to be good, however it is best to be receptive to suggestions.

The abstract is perhaps the only thing that manuscript authors read, what is the need to put data that does not focus on the research question or the title of the manuscript?

The second paragraph talks about giardiasis as a factor related to multiple problems, what does that have to do with genomic characterization of giardia? This space should focus on the characteristics of assemblages, the difficulties of microscopic recognition, the advantages of molecular testing, etc., all directed to your work.

Table 1 mentions genetic diversity and locates papers that have not assessed this variable, what is the need?

The ethical PROCEDURES followed in the work should normally be mentioned. Was each subject coded? How was anonymization performed? Who had access to personal data? Remembering that the research subjects are children. That goes beyond the ethical statement at the end and something so important cannot be on request, it must be mentioned in the manuscript.

The supplementary tables are not, and remain unavailable.

The manuscript is not authored by me, or even my co-authors, I suggested it because it seemed relevant.

You try to cover big things in your manuscript and stop focusing on the main reason for your work.

Author Response

The sole purpose of making the suggestions was to improve the content and focus of your manuscript. Unfortunately, like every other author including myself, we consider our work to be good, however it is best to be receptive to suggestions.

Reply: please rest assured that we have seriously considered all the comments and suggestions raised by all the external reviewers that appraised our work. In those cases where we decided not to consider them, we provided a through explanation for that decision. As Reviewer #2 pointed out correctly, adopting an open-mind frame is essential when dealing with scientific criticism. By no means this implies that authors must accept all comments/suggestions proposed by reviewers if they have a strong case not to do so.

  1. The abstract is perhaps the only thing that manuscript authors read, what is the need to put data that does not focus on the research question or the title of the manuscript?

Reply: As already discussed in the first round of revision, this is precisely our point: not adding this information in the Abstract section would result in losing it for interested readers. This is the main reason why we wish to keep it as it stands now.

  1. The second paragraph talks about giardiasis as a factor related to multiple problems, what does that have to do with genomic characterization of Giardia? This space should focus on the characteristics of assemblages, the difficulties of microscopic recognition, the advantages of molecular testing, etc., all directed to your work.

Reply: we feel that Reviewer #2 is truly missing the point and justification of this survey. In our opinion, we have already provided a robust and thorough explanation for this point in our answer to comment #2 by Reviewer #2 in the first round of revision. We do not have further comments on this matter.

  1. Table 1 mentions genetic diversity and locates papers that have not assessed this variable, what is the need?

Reply: we are puzzled of still receiving this comment, particularly considering our thorough response to comment #2 by Reviewer #2 in the first round of revision. We do not have further comments on this matter.

  1. The ethical PROCEDURES followed in the work should normally be mentioned. Was each subject coded? How was anonymization performed? Who had access to personal data? Remembering that the research subjects are children. That goes beyond the ethical statement at the end and something so important cannot be on request, it must be mentioned in the manuscript.

Reply: in an attempt to satisfy Reviewer#2, we have added the following information in current lines xx-xx of the main body of the manuscript: “To protect the privacy of the participants and their information from unauthorized access, samples and epidemiological questionnaires were anonymized by assigning them a numeric code. Only the principal investigator of the research project (M.K.T.) had access to this sensitive in-formation”.

  1. The supplementary tables are not, and remain unavailable.

Reply: we assure Reviewer #2 that Supplementary Tables were part of the two previous submissions. We do not understand why Reviewer #2 cannot access them. To the best of our knowledge, none of the other five external reviewers that appraised the work have experienced this problem.

  1. The manuscript is not authored by me, or even my co-authors, I suggested it because it seemed relevant.

Reply: we already discussed this point in our thorough reply to comment #6 by Reviewer #2 in the first round of revision. We do not have further comments on this matter.

  1. You try to cover big things in your manuscript and stop focusing on the main reason for your work.

Reply: We thank Reviewer #2 for his/her advice and will take it into consideration in our future work.

Reviewer 3 Report

The authors did not accept all reviewers comments. If it is only focused in Nigeria they should submit it to a regional journal. 

Author Response

The authors did not accept all reviewers comments. If it is only focused in Nigeria they should submit it to a regional journal.

Reply: please see our detailed answer to the first comment by Reviewer #2 in this round of revision. We do not have further comments on this matter.